# Glass as a Fine Art Medium: Brief History and the Role of Adriano Berengo as a Fine Art Glass Impresario

**Goshka Bialek**

Independent Researcher, Durham DH1 4ED, UK; goshkab@gmail.com

**Abstract:** This article explores the role of glass as a medium in the fine arts rather than as a craft form. It includes a short history of glass as an art medium, the development of glass technologies and their application in the field of fine art. It reflects the distinctiveness of glass as a sculptural medium due to its optical properties and transparency; glass's inherent characteristics create the unique possibility of using the space both outside and inside a solid object. This article, furthermore, demonstrates the importance of specific individuals in bringing glass as a fine art medium to the fore, in particular Adriano Berengo. Berengo proves exceptional in promoting glass in the field of fine arts and has been particularly effective in encouraging well-known artists to experiment with it as a medium. The article discusses the impact of his efforts to establish cooperation with great names from all over the world, from Ai Weiwei to Tony Cragg and from Jaume Plensa to César, who have passed through Adriano Berengo's studio.

**Keywords:** fine arts; sculpture; glass; glass art; Glasstress; inner space; crafts; arts medium

## 1. Introduction

The United Nations declared the year 2022 as the International Year of Glass (McDonald 2021). This presents an exciting opportunity to explore the uniqueness of glass as a fine art medium, why it has historically not been a popular medium and the possibilities it holds for the fine art world going forwards.

We can already see the impact of the UN declaration, with Ireland already announcing an artist who uses glass as a medium, Niamh O'Malley, to represent them at the Venice Biennale (Niamh O'Malley to Represent Ireland at Venice Biennale 2021). This mainstreaming of glass within fine art represents an important step towards recognising the power that it can provide as a medium.

The history of artistic glasswork and industrial glass production is full of challenges overcome and challenges that, through creativity, have led to new opportunities. The experiences of industrial glass producers are important to artists: learnings from these highly technological environments can be applied creatively to glasswork in studios especially when sat beside partnerships with industry, which facilitate access to specialist (and expensive) equipment (Bialek 2017). We are currently seeing a growth in partnerships between artists and glass technologists which is creating exciting results; however, it is important that this is not at the expense of artists working with glass themselves as experimenting with a material often leads to important creative (and technological) leaps forward.

An increasing number of artists are trying to use glass as a medium to create their sculptures, to a large extent, due to the optical properties and transparency of this material. However, usually those artists tended to use factory-made glass with glass casting or hot glass forming techniques being used much less frequently in fine arts. As technology evolves, we encounter fewer restrictions. However, restrictions on size remain a significant constraint.

Access to industrial technologies is still primarily only available to artists working in or associated with glass factories. In Scandinavia, glass art is still closely linked to the glass

industry and this offers the opportunity to apply some of the more expensive technologies (Dowson 2000). One of these artists is Bertil Vallien. He started to work with glass when he began to work for Åfors glass factory as a designer in 1963. Glass started to be his most inspirational medium with the huge development of sand-casting methods (Figure 1). Thanks to this method, he is less restricted by size than many glass artists. Some of his sculptures are up to 14 feet long (Giubilei 2012). His sand-casting process gave him the opportunity to use the medium in a more personal way without the restrictions which other available materials, tools and processes offered (Vallien 2013).

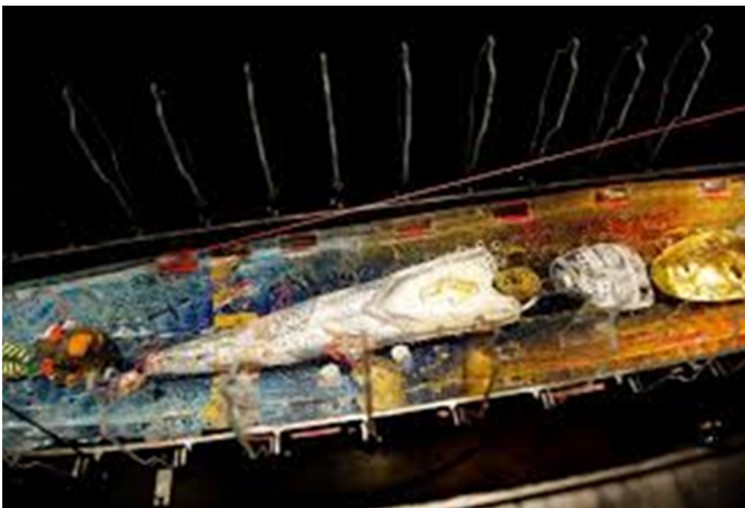

**Figure 1.** Bertil Vallien; Boat; 2016. Made for American collectors on 17 May 2016. Photo: Lena Gunnarsson. Used with permission.

However, many artists prefer to work and create in their studios. The second half of the 20th century saw a great interest in moving the use of glass as a medium beyond craft and into fine art. There was significant development of new technologies and their application in sculpture in artists' studios.

In the 1940's and 1950's Czech artists developed studio-based fine art glass to a very high level, but until the 1960's they could not share their achievements freely due to political constraints. Czech artist started enjoying international recognition for their glass artworks such as sculptures when it was displayed and awarded in international exhibitions in Expo 58 World Fair in Brussels and in Expo 67 in Montréal (Petrova 2001). Nevertheless, Bohemia (a part of the Czech Republic) became famous for its glass art already during the Renaissance, and the oldest archaeological excavations of glass production date even to around 1250 (Petrova 2001).

Separately, 1950's American artists interested in glass began to look for new ways of working with this material outside of industry. Harvey K. Littleton, a teaching ceramist, started experimenting successfully with hot glass. In the 1960's he collaborated with glass research scientist Dominick Labino, who built a small and affordable furnace for melting glass that could be used in artists' studios (Klein 1989). In 1962, Littleton delivered two glass-blowing workshops at the Toledo Museum of Art to introduce artists to the use of hot glass in the studio. This was the beginning of the American Studio Glass movement.

As American Studio glass artists felt the lack of technical knowledge to develop truly sculptural objects, they sought help in countries with more experience in this field such as Czechoslovakia, Sweden, and Italy. This led to the Studio Glass movement quickly spreading to Europe and then gradually to the other continents. A characteristic of this movement was the sharing of technical knowledge and ideas between artists, which is not possible within the glass industry due to commercial constraints. However, the studio movement placed emphasis on artists as designers and makers, which had a major influence on limiting the use of glass in the fine arts.

This limitation was noted by Adriano Berengo (Born 1947 Venice, Italy), perhaps the most important figure in enabling artists to work in glass and promoting glass as a fine art medium (Berengo 2021). Berengo is a gallerist, a collector, a dealer, an entrepreneur and most importantly, an open-minded visionary. One of his ideas was to invite artists from all over the world to work with his studio (Figures 2 and 11); he introduced the potential of glass as a medium to them; facilitated collaborations between them and glass masters to help to create their projects, and, as a result, supported the recognition of glass as a distinctive medium within fine art.

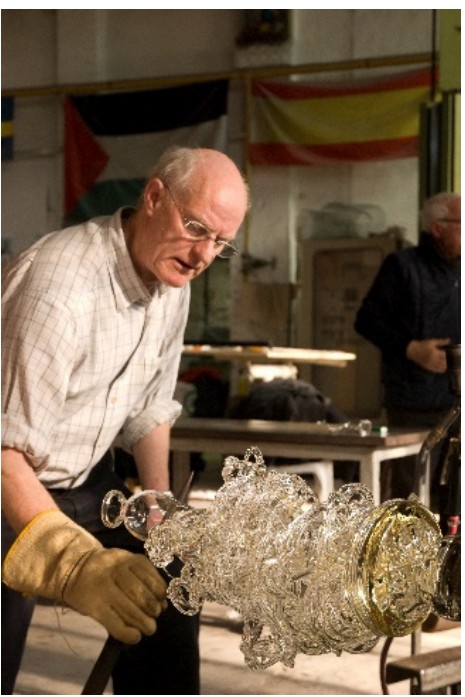

**Figure 2.** Tony Cragg in Berengo Studio 2009. Photographed by Francesco Allegretto; used with permission.

Berengo developed the Glasstress project to further his mission that glass be recognised as an important part of contemporary art by introducing the Berengo Studio glass creations at the Venice Biennale for the first time in 2009 and has successfully continued it to this day. In addition, to increase the international impact of Glasstress, Berengo organised these exhibitions in cooperation with prestigious museums including the State Hermitage Museum in St. Petersburg (Russia), the Museum of Art and Design in New York and the Millesgården Museum in Stockholm. In order to show objectively the development of the use of glass in the fine arts, he included works by well-known artists in his exhibitions, even though they were made by other fabricators.

In this article we will explore why glass, unlike many other materials, has not been fully recognised as a fine art medium throughout history: to do so we will present a brief history of glasswork, with emphasis on its applications in art and Adriano Berengo's influence on the position of glass art as a contemporary fine art.

## 2. Short History of Glass as an Art Medium

The mediums of sculpture have, through the centuries, been subject to a system of hierarchies based on social, political and aesthetic aspects. The oldest three materials applied in sculpture are clay, marble and bronze. Glass, like clay, is one of the oldest media in existence. Glass has been known to humans since they started to use tools. During the Stone Age (from 3.4 million years BCE to 8000 BCE), people made tools and weapons from natural volcanic glass. However, most historians of glass are agreed that techniques for making glass were first discovered in the Bronze Age around the end of the third

millennium BCE. At this point in time, the process of melting and casting metal was already known, and this knowledge proved useful in the production of glass. Humans have been producing glass by melting raw materials for thousands of years. First beads, which were supposed to be substitutes for semiprecious and precious stones, were the most common glass products, but glass rods, inlays and other richly coloured items were also produced (Grose 1989).

Venetian glass techniques (predominantly based on freehand glass blowing) have been known since the 8th century CE. In the 13th century, glass workshops moved to Murano, to protect Venice from the risk of fire, and also to maintain the secrecy surrounding the techniques of glass blowing (and therefore retain the commercial advantage that this offered (Gable 2004). In the 15th and 16th centuries, Murano glass was in the full bloom of glass making and development of unique techniques, but in the 17th century, it entered the period of gradual decline, which was continued to the 19th century. In 1895 the first Venice Biennial Exhibition was introduced, and it highlighted the gap between the modern trends in Europe and Murano artisan production in styles from the past (Toso 1999). At the beginning of the 20th century, Murano returned to art and continuous innovation, which led to a rise in popularity (Gable 2004).

In the 1950's, Egidio Costantini (1912, Brindisi, Italy–2007, Venice, Italy) had a vision of how to promote Murano glass more widely, moving it beyond a craft material and into the world of fine art. He was fascinated by Murano glass but was not a master glassmaker himself. His idea was to increase access to glass workshops for artists by producing glass sculptures from drawings by well-known contemporary artists such as Pablo Picasso (Figure 3), Marc Chagall and Jean Arp (Peggy Guggenheim Collection 2021). By associating the work of the workshops with famous artists, he planned to elevate the art of glassblowing to the same level as sculpture or painting. It was a highly innovative proposal as outsiders did not have access to the Murano workshops at that time. He built a collaboration of Venetian artists and set up the Centro Studio Pittori nell'Arte del Vetro di Murano. However, this only lasted five years. Following this, he opened his own gallery, the Fucina degli Angeli, which was successful reputationally but less so financially. Peggy Guggenheim became involved as a funder of the gallery as it was also important to her that Venetian art and Murano glass art be recognised worldwide. The last exhibition of Constantini works was held in Innsbruck in 2003, although pieces based created from drawings by famous artists are still on display at the Peggy Guggenheim Museum.

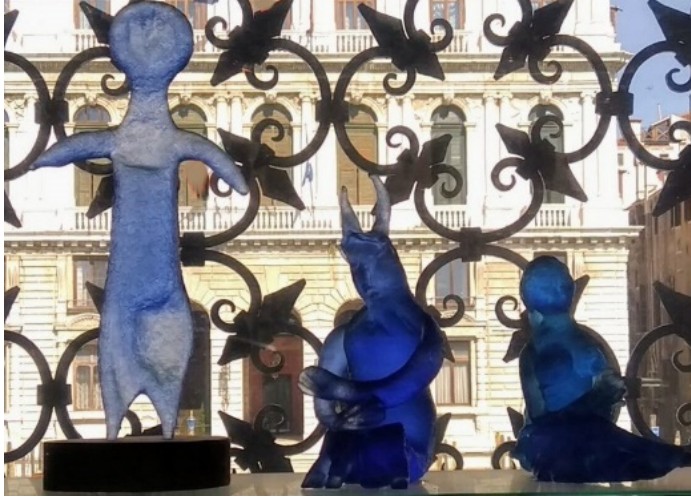

**Figure 3.** Glass sculptures created by Egidio Costantini from Picasso's drawings. 1964; Peggy Guggenheim Collection in Venice. Photographed by Goshka Bialek.

For a long time, Murano was at the forefront of glass production, but this does not mean that they were the only ones on the world stage. For example, Bohemian glass, produced in the regions of Bohemia and Silesia (now parts of the Czech Republic) from around 1600 was internationally recognised for craftsmanship and innovative designs.

At the end of the 19th century and the beginning of the 20th century, the development of mass production of glass slowed down the progress of small glass craft manufacture, which was the driving force in the development of new techniques in the production of glass (Cousin 1996). However, one of the exceptions was the development of French glass. French glass factories such as Baccarat and St. Louis were founded in the 1760's. During the 19th and early 20th centuries, French glassmakers became world leaders, with artists such as Lalique, Galle and Schneider.

Nevertheless, until the early 20th-century glass was treated as a craft material. This was almost certainly due to technological constraints, cost and limited access to glass workshops for artists. Scientific developments and the Industrial Revolution led to the development of new technologies and materials and lower production costs, all making it more accessible to artists as a material.

The pioneers in the use of the glass for a conceptual sculpture were two brothers, the Russian Constructivists: Antoine Pevsner and Naum Gabo (previously known as Naum Neemia Pevsner). The Constructivists promoted the use of contemporary industrial materials; they did not carve or model these materials according to sculptural conventions but constructed them according to principles of modern technology.

The two brothers were some of the several Constructivist artists involved in the Bauhaus movement, linked by a common approach to art development. The Bauhaus philosophy was: "Architects, sculptors, painters, we must all turn to the crafts" (Bayer Gropius, p. 18). Sculpture was taught alongside other art disciplines where an important part was played by instruction in materials and tools. Between 1917 and 1924, the brothers experimented with the principles of constructive form and new materials (in the field of fine art) like glass and plastic were used for the first time (Figure 4) to articulate a new sense of space (Read 1998, p. 110), and this type of glass application has continued into present times, new methods have been developed and used by artists such as Marcel Duchamp, Larry Bell and Gerhard Richter.

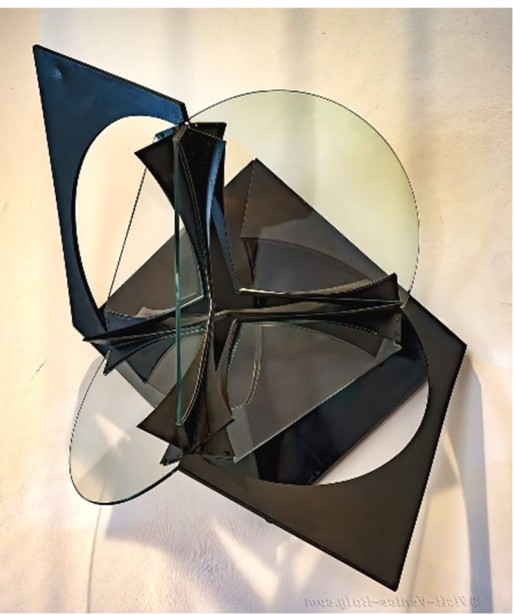

**Figure 4.** Antoine Pevsner; Anchored Cross; 1925. Peggy Guggenheim Museum in Venice; photographed by Goshka Bialek. Used with permission.

The 1930s were years of expansion and consolidation in modernist sculpture. Most sculptors who worked with stone used the process of carving to bring the stone itself to life (Clough 1969). Barbara Hepworth innovated with this methodology, making a hole in a solid object made from stone, this new space turning into its own form (Hammacher 1987, p. 40). The hole connects one side sculpture to the other, making it instantly more three dimensional. Seeing both sides at the same time is impossible unless the sculpture is created from a transparent medium such as glass but it was not until the 1960s that Hepworth experimented with hot glass, using it in Four Hemispheres (although the sculpture was executed by a local glass factory). It continued a long line of thematic exploration within her artistic practice—the hemisphere first appearing in works such as the 1937 marble Pierced Hemisphere I, The Hepworth Wakefield.

In the first half of the 20th century many artists around the world, often in conjunction with technologists, tried to adapt glass to expand its use in architecture and art. However, the adoption of hot glass techniques in art required a lot of technological knowledge and appropriate equipment. Murano masters and others were not willing to share their knowledge, so the adaptation of these techniques in the fine arts, and especially in artists' studios, took a little longer. One of the first artists who were interested in developing the possibilities of hot glass techniques in their artistic practice was Vera Mukhina.

Vera Mukhina (Мухина Вера Игнатьевна; born: 1 July 1889, Riga, Latvia, died: 6 October 1953, Moscow, Russia), was a prominent Soviet multidisciplinary artist, internationally recognised for her monumental work Worker and Kolkhoz Woman for the 1937 World's Fair in Paris. She was also interested in incorporating glass in her artistic practice and was one of the pioneers of Soviet glass art and glassware sets which were made from her sketches (Dzandzugazova 2013).

During a visit to Italy, in particular the Murano workshops, in 1914 she became interested in glass. However, she began to work with this new medium at the beginning of 40's when she, along with Nikolai Kachalov (inventor of optic glass technology) gained permission to open a studio for mass production of glassware and funding for an experimental laboratory. After the war, Vera Mukhina, under the instruction of Nikolai Kachalov, began to apply glass to her artistic practice. She started with small forms but would eventually begin using her own hot glass techniques to produce sculptures (Figure 5).

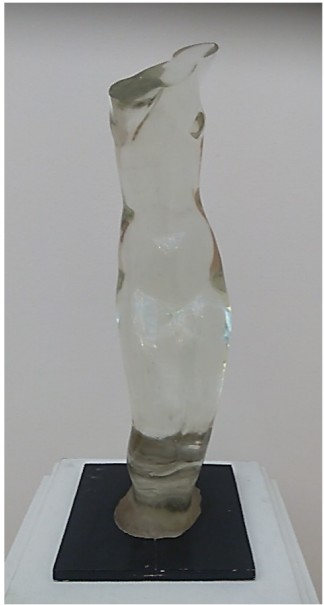

**Figure 5.** Vera Mukhina; Female torso (Женский торс); 1947, New Tretyakov Gallery, Moscow; photographed by Goshka Bialek.

In 1962, as was mentioned earlier, workshops led by Harvey Littleton were held in The Toledo Museum of Art in Ohio, USA, to explore the possibilities of melting glass in artists' studios (Cummings 2002) sparking the American Studio Glass movement. Following the success of these workshops, an international series of workshops and conferences were organised for artists, curators and technologists, which addressed the different applications of glass, as well as combining glass with other materials. This period brought about a dramatic transformation in glass art and the development of glass studios run by artists themselves (reducing their dependence on specialist workshops).

At the same time as the blossoming of the American Studio Glass movement, Czech glass artists finally became able to share their art on the international stage; this also proved an important turning point in the treatment of glass as a fine art medium.

Further development pushing the boundaries of glass as a fine art medium took place in the 1970s (Dorigato and Klein 1996). In the introduction to the catalogue from the first major international New Glass exhibition organised by the Corning Museum in 1979, the founding director of the Corning Museum of Glass, Thomas Buechner, stated that a dramatic change was taking place in the history of glass, after thirty-five centuries of utilitarian use of this medium (Klein 1989).

However, the limitations of the size of glass sculptures did not encourage artists, especially not technologically skilled artists, to use the medium. This led to the development of art fabricators working with glass, who were often associated with art or were artists themselves, as Kunstgiesserei St. Gallen founded by Felix Lehner in Switzerland, Cirva located in Marseille in France or Zdeněk Lhotský Studio in the Czech Republic. Zdenek Lhotsky is a painter, sculptor and founding member of the artistic group The Stubborn Ones. Additionally, he is a successful glass fabricator. He cast monumental works in glass designed by himself and other artists. For example, he developed technology to execute Karen Lamonte glass figures, which she had designed in wax.

In recent years, the art fabricators' manufacturing costs in Europe, the USA and Australia have become uncompetitive, with the result that more and more glass fabricators are opening and/or operating in China. This poses a huge challenge, especially for the European artist industry.

Apart from that, a common problem for artists can be the difficulty of communication between artists and fabricators, who have the technical skills but insufficient understanding of fine arts. Many famous artists solve this by setting up their own studios. One of them is Olafur Eliasson. In 1995, he moved to Berlin and founded Studio Olafur Eliasson, which today comprises a large team of craftsmen, architects, archivists, researchers, administrators, cooks, programmers, art historians and specialised technicians. The Studio Glass movement has been characterised by the exchange of ideas between artists and provides artists with possibilities to explore applications of glass. In addition, the development of the internet and social media has led to better communication, sharing of information and increased collaboration between participants. Numerous social media groups have emerged to share experiences and information and to promote the activities of artists, using glass in their artistic practice, to a wider audience (for example, 21st Century Glass: Conversations and Images/Glass Secessionism created by Tim Tate and William Warmus). Most of the groups concentrate on particular materials used and the techniques employed in the application, and members can discuss specific problems related to the subject of the forum.

Despite the wide exchange of ideas and knowledge, these forums primarily bring together glass artists, leaving a continuing divide between glass artists and fine artists adopting glass as a medium. This risks a return to the viewpoint of glass as craft rather than glass as fine art.

### 3. Glass as a Medium

From the point of view of technical applications, the most favourable properties of glass include substantial chemical resistance, good dielectric properties, high light transmittance, low thermal conductivity, high hardness, low abrasion and ease of shaping

and obtaining a smooth surface (Nowotny 1963). From an artistic point of view, the most interesting of these are transparency, reflectiveness and high light transmittance, because these features enable interesting optical effects and enrich the possibilities of visual manipulation.

However, the mechanical properties of glass make it a challenging material to use. Selection of the appropriate type of glass (a highly technical process in its own right) significantly improves how easily it can be used (Zieba 1987); however, even with the correct selection and a high level of technical skill, the end result is not guaranteed, especially with monumental works. There continue to be significant limitations of the size of glass artworks; in the case of blown glass the limitation is the weight that a person can carry, and, in the case of cast glass the limitation is the size of kilns and the capacity to anneal (cool) the glass in a controlled way (this becomes progressively more challenging as the glass becomes thicker).

The aesthetic properties of glass as reflectiveness and transparency raise unique sculptural issues. There are vast differences between sculptures in glass and in other media due to their unique optical properties—particularly the ability to create and use interior "spaces" which are visible from the exterior. It is necessary to consider internal space in sculpture that arise because of the transparency of the material used. If an object is opaque, the only part that we can see is the boundary between the space occupied by the object and the unoccupied space (Bialek 2017). In this case, the object defines a shape in space by delimiting it. It is the sculpture's facade behind which we assume space hides, but we are not able to see further in (George 2014). Space is one of the most important parts of form and it is the space we need to see. This is illustrated by Libensky and Brychtova, leading glass artists, being obliged to invent new phrases like "colourful glass space", "coloured object in space", "space light dimension" or "light colour" to discuss their work. These phrases were critical to creating descriptions of the carefully considered hollow interior cavities they create in their work that defines the colour and luminosity (Kehlmann 2001). The inner space itself, therefore, becomes a new and important medium in sculpture. However, it is not space itself that becomes visible, only its absence. By using glass, it is possible to make space inside sculpture visible and solid, and to create new forms within this space (Bialek 2017). Glass artists such as Bertil Vallien use space inside their sculptures. Indeed, Vallien is "more interested in what goes on inside the glass than . . . in the outer shell" (Vallien 2013).

The properties of glass can generate a great deal of interest among fine artists and expand the possibilities for glass to be used extensively in the fine arts, provided that artists can have easier access to working with the material and that the potential of glass as a medium is adequately promoted. Development in this direction has been noticeable in recent years.

## 4. Adriano Berengo and His Projects to Elevated Importance of Glass as Art Medium

There have been several attempts in the past to present glass as a fine art material and to have it consistently accepted as such: most notably, the Egidio Costantini–Peggy Guggenheim collaboration which introduced famous international artists such as Pablo Picasso, Jean Arp and Max Ernst to glass as a fine art medium.

Inspired by this, Adriano Berengo took action to elevate the importance of glass as a fine art medium. He has actively promoted knowledge of glass as a fine art material and provided the opportunity to work with it to a wide group of international artists, many of them have no previous experience with glass (Figures 2 and 11), alongside presenting the glass artworks created to a wide international audience at high-profile international events.

In 1989 Adriano Berengo established the Berengo Studio. The main objective of Adriano Berengo's work is to bring glass into the world of fine art and as such into the international arena. Adriano invites artists, architects and designers from different countries, known and less known, to create an innovative application of glass. He is also interested in people for whom glass is a new medium in their artistic practice. He is also

interested in new ideas for the promotion of glass, as well as new technologies for the use of glass and, if he sees potential, he allows artists to carry out trials in his studio (Hawlin 2020).

As part of its activities, Berengo Studio initiated the Glasstress project in 2009 (as an official collateral event of the Venice Biennale) to represent the possibilities of glass as a material for contemporary art (Glasstress 2009). The project mission was to introduce the Berengo Studio glass creations and also to show how contemporary artists apply the medium to a wider public. The shape of the project has changed over the years. The first four Glasstress exhibitions were official events in the Venice Biennale, with each exhibition bringing a new perspective on glass.

Glasstress Gotika 2015 was organised by the State Hermitage Museum in conjunction with Berengo Studio in the Palazzo Cavalli-Franchetti in Venice. The exhibition presented the Gothic and neo-Gothic objects from the museum collection in dialogue with works created by contemporary artists, some of them at the Berengo Studio on Murano. The exhibition included artworks from Olafur Eliasson, Erwin Wurm, Jaume Plensa, Tony Cragg and the brothers Chapman (Glasstress 2015).

Glasstress 2017 evolved from the previous events and was an independent exhibition that took place during the Venice Biennale of Art. After founding Fondazione Berengo in 2014, Berengo opened what is now the Fondazione Berengo Art Space in an old unused glass factory in Murano, which became the venue for Glasstress 2019 (Berengo 2018). Additionally, the events turned into a travelling exhibition, not merely held in Venice but held in venues all over the world.

Under invitation from Adriano Berengo, more than 300 artists, architects, and designers have been given the opportunity to creatively express their thoughts and ideas using glass. This has led to more and more contemporary sculptors and conceptual artists working with glass, such as Ai Weiwei (Figure 6), Sir Tony Cragg (Figure 7), Jaume Plensa, the Chapman Brothers, Mona Hatoum (Figure 8) and Cesar and Miroslaw Balka.

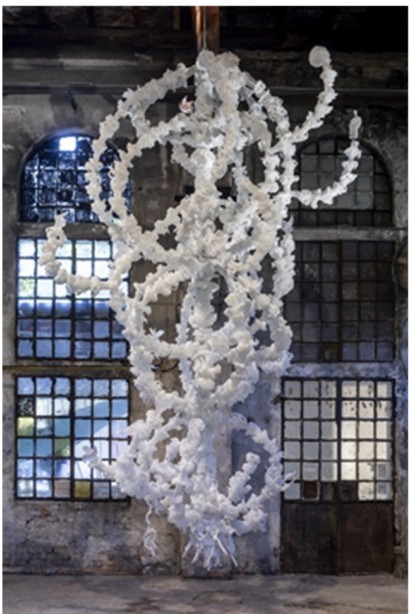

**Figure 6.** Ai Weiwei's Blossom 2017; white glass chandelier—looping flowers complete with security cameras; Berengo Studio. Used with permission.

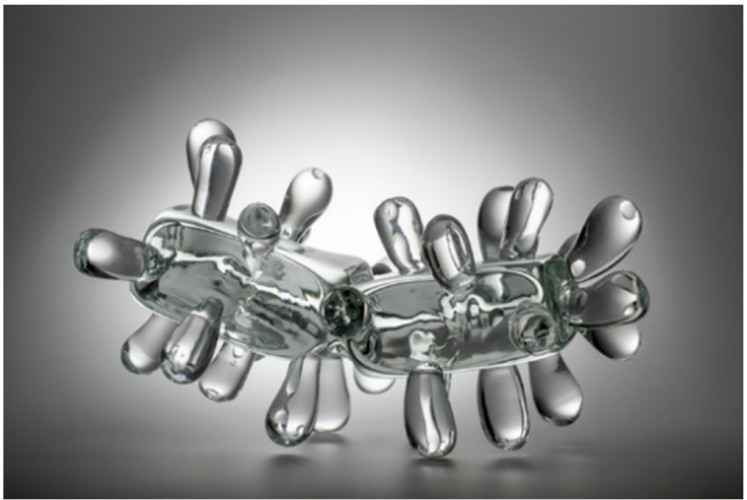

**Figure 7.** Sir Tony Cragg, Untitled, 2015; hot glass; Berengo Studio; Glasstress 2015. Used with permission.

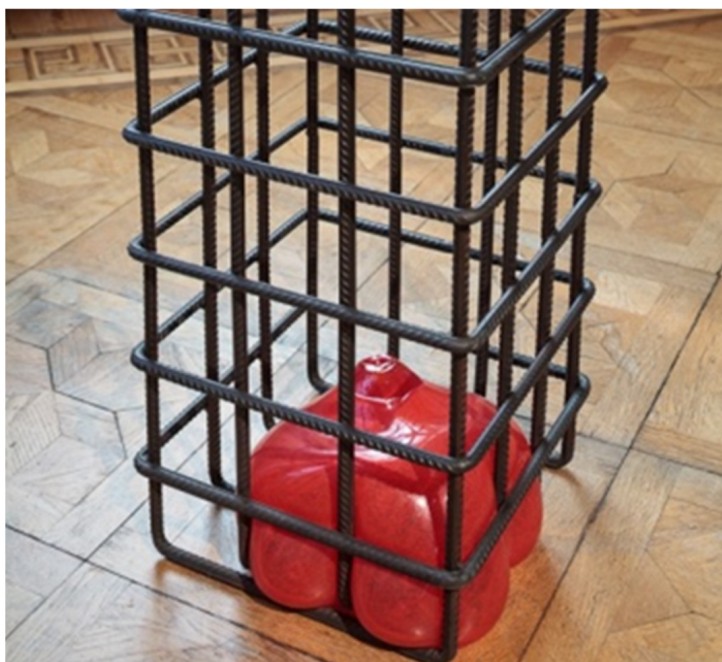

**Figure 8.** Mona Hatoum; Kapancik, 2012; metal and glass combination; Glasstress 2013. Used with permission.

Mona Hatoum carried out her first attempts to create the Kapancik sculpture with the assistance of Adriano Berengo masters, and she continued her glass application in other studios (Berengo 2021). Adriano Berengo's idea is to present and promote the widest possible use of glass in the fine arts, which is why he exhibits the work of famous artists who have not necessarily worked in his studio: Louise Bourgeois (Figure 9) or Olafur Eliasson (Figure 10).

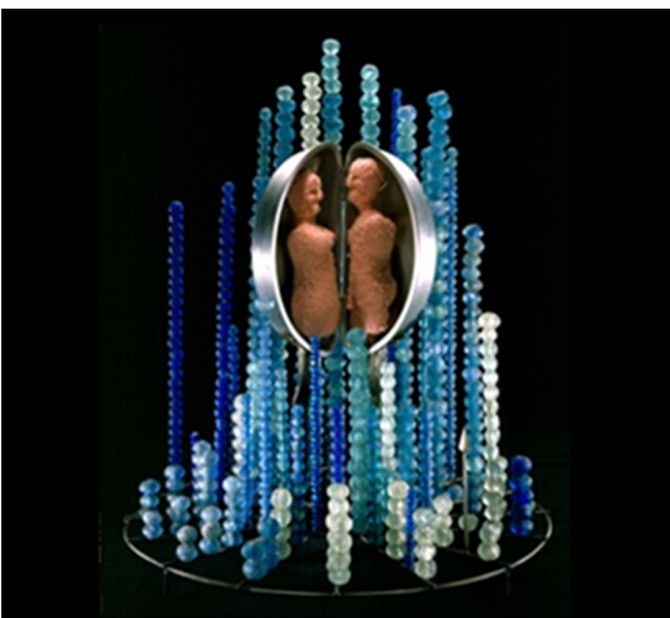

**Figure 9.** Louise Bourgeois, The Couple, 2002; glass and metal combination; Glasstress 2009. Used with permission.

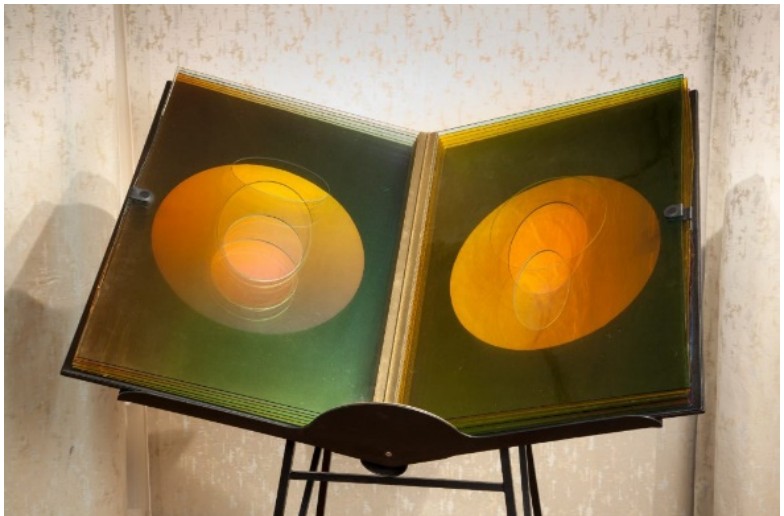

**Figure 10.** Olafur Eliasson; A View Becomes a Window, 2013; glass, leather; Glasstress 2015. Used with permission.

The most recent exhibition "Glasstress Window To The Future" (11 September–31 October 2021), in the State Hermitage, St Petersburg, Russia featured works by 51 artists, including key figures in contemporary art such as Ai Weiwei, Renate Bertlmann, Koen Vanmechelen, Michael Joo, Ilya and Emilia Kabakov and Sean Scully (Figure 11). The exhibition has been prepared by the Hermitage in conjunction with the Berengo Studio of Venice (Berengo 2021).

Adriano Berengo introduced Ai Weiwei to glass as a medium (Figure 12) and since then he has become fascinated by it. Like Berengo he "believes in contemporary expression, but at the same time tries to develop this old technique into a new language" (Ai Weiwei, Fondazioneberengo.org; (Berengo 2021).

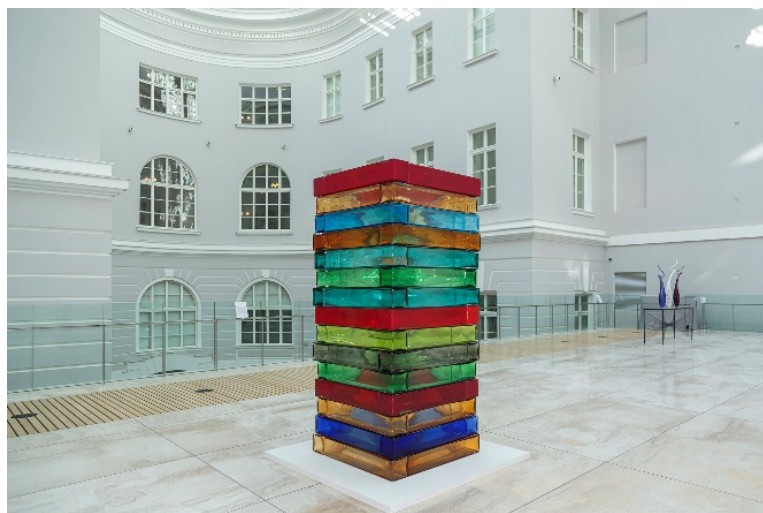

**Figure 11.** Sean Scully; The question is are you wrapped rigidly in the cloth of your time or can you fly out of it? 2021; glass; Glasstress Window to the Future; the State Hermitage, St Petersburg, Russia; photographed by Svetlana Ragina. Used with permission.

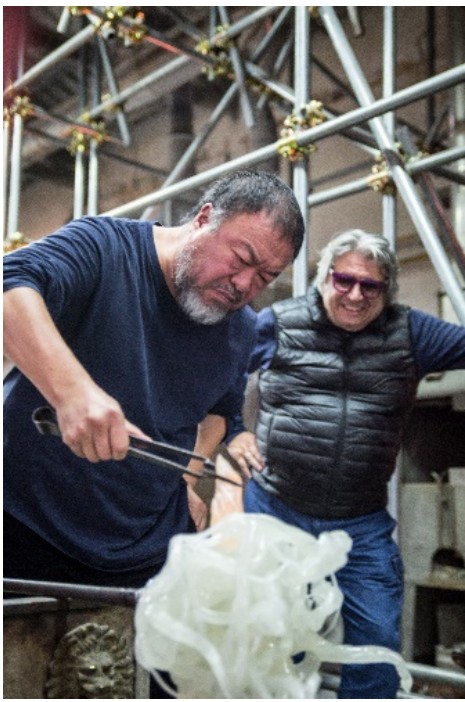

**Figure 12.** Ai Weiwei with Adriano Berengo in the Fondazione Berengo Art Space Murano 2016. Courtesy: © Karolina Sobel. Used with permission.

Ai Weiwei has extended his use of glass beyond Glasstress and into his general creative practice. Examples of this include his exhibition Cubes and Trees (Downing College, Cambridge 2016) and the piece Crystal Cube (2014).

Cubes and Trees, an exhibition that combined sculpture with other forms such as video, focused on two series of sculptures: a series of trees created from discarded pieces of wood and a series of cubes made from compacted tea, carved ebony, huali wood and crystal glass that relate minimalist sculptural forms with Chinese craft and heritage.

Crystal Cube (Figure 13) as part of the installation was cast in China and was exhibited at Art Basel, Miami (2014) with Ai Weiwei's photo, which was taken during his arrest, in

an elevator with the arresting policemen. This work is technically interesting, as well as conceptually, as it is one of the largest successful sculptures in cast glass (Warmus 2016).

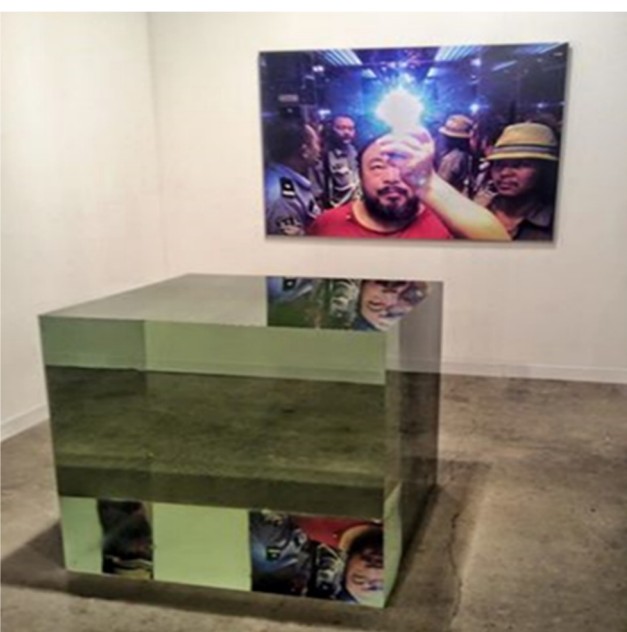

**Figure 13.** Ai Weiwei; Crystal Cube; 2014; Art Basel; crystal (leaded glass) cube 100 × 100 × 100 cm (1 cubic meter). Photographed by William Warmus. Used with permission.

Ai Weiwei is a contemporary artist, but he is " . . . always learning and working with tradition" (Ai Weiwei, Fondazioneberengo.org) and as such glass offers him a significant opportunity. He continues to be a frequent visitor to Berengo's studio and together they work on new challenging projects using glass.

With the increased promotion of glass as a fine art medium and the reduction in technical constraints of working with it (although size remains a continuing limitation) we see an increasing number of artists taking advantage of the possibilities that it offers aesthetically and conceptually.

Ai Weiwei's, and other respected contemporary artists', recognition of glass as a powerful medium of expression will continue to be critical in bridging the gap between contemporary glass and contemporary fine art—cementing glass' place as a fine art medium.

## 5. Conclusions

In recent years, there has been considerable interest for sculptors in the creative potential of a glass medium. Unlike other fine art materials that are much more widespread because they are easier to use, glass is a very difficult medium from a technological point and requires both technical skills and knowledge. Sculptors find it difficult to employ this material for the development of artistic ideas because skills and knowledge are acquired only through many years of experience working directly with this medium.

Thanks to the Berengo Studio's activities and successful Glasstress project many new aspects have come to enrich artists' way of viewing glass as a fine art medium. Besides meticulously designed objects, realised with the highest precision, works with a more conceptual approach are becoming increasingly important. New solutions for turning creative ideas into reality are opening, made possible by new production and processing techniques.

Adriano Berengo's innovative approach and highly visionary ideas have led to glass being recognised and elevated as an important medium in art. The difficult global economic situation, which was not helped by the COVID-19 problems, did not stop him in his tracks, and his high standard of numerous exhibitions and projects will certainly enrich the

planned activities on the occasion of the International Year of Glass 2022 announced by the United Nations.

**Funding:** This research received no external funding.

**Institutional Review Board Statement:** The study did not require ethical approval.

**Informed Consent Statement:** Not applicable.

**Data Availability Statement:** Did not report any data.

**Conflicts of Interest:** The authors declare no conflict of interest.

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
