# Peer review of "Glass as a Fine Art Medium: Brief History and the Role of Adriano Berengo as a Fine Art Glass Impresario"

_arts, 2021_

Round 1

Reviewer 1 Report

This article aims to explore the use of glass as a fine art medium. As glass is a highly technical medium that requires expertise and expensive facilities, the article highlights one individual, Adriano Berengo, who has been initiating and facilitating the use of glass as a fine art medium in his studio.

The manuscript is clear and relevant to the field. While the cited references are mostly considerably older than 5 years, they are still valid, especially where the history of glassmaking is concerned. However, facts should be double-checked (see specific comments below). The structure seems slightly opaque to me – there are two distinct sections about the history of glass – line 57-78 and 100-206. Could this be brought together?

There are some gaps in this research; the conclusion is essentially accurate, but brief and somewhat one-sided. There are several well-known fine artists who work with glass without facilitation by Berengo, such as Larry Bell, Roni Horn and Olafur Eliason and, of course, Marcel Duchamp. They should be referred to. (Eliason is mentioned as exhibiting in Glasstress Gotika 2015, but he has been working with glass since the early 90s). While the article briefly touches on the crafts/fine art divide, there is no mention of the challenges Berengo studio  faces in facilitating artwork in a new (for the artists), and technically difficult, medium.

65-70 not quite accurate: Harvey Littleton applied for and received funding to build a small furnace at the Toledo museum of art, not in his studio. There, he experimented with 5 of his graduate students. The author mentions this, without naming Littleton, in lines 190-192. This should be updated and brought together. https://www.cmog.org/article/harvey-k-littleton-and-american-studio-glass-movement

line 203-204: “the current President of the Corning Museum, Thomas Buechner” – This is  inaccurate: Thomas Buechner was the founding director of the Corning Museum of Glass, later president of Steuben Glass and the Rockwell Museum, but never president of the Corning Museum of glass. He died in 2010. https://www.cmog.org/bio/thomas-s-buechner

Author Response

I would like to thank for the Reviewer’s thoughtful comments that have allowed me to improve my article. Below are my replies (in red) to specific points raised. The modified parts of the article are indicated in blue.

This article aims to explore the use of glass as a fine art medium. As glass is a highly technical medium that requires expertise and expensive facilities, the article highlights one individual, Adriano Berengo, who has been initiating and facilitating the use of glass as a fine art medium in his studio.

In my opinion, Berengo is very unique and his activities cover most of the areas to be successful with glass as the fine art medium. As part of its activities, is not only facilitating the artist, it is also the spreading knowledge about glass, to show how contemporary artists apply the medium to a wider public by initiated the Glasstress project (as an official collateral event of the Venice Biennale) to represent the possibilities of glass as a material for contemporary art. I am writing about this through my article

The manuscript is clear and relevant to the field. While the cited references are mostly considerably older than 5 years, they are still valid, especially where the history of glassmaking is concerned. However, facts should be double-checked (see specific comments below). The structure seems slightly opaque to me – there are two distinct sections about the history of glass – line 57-78  and 100-206. Could this be brought together?

I can see the point the Reviewer makes but, after careful consideration, I have decided to keep the structure as it is. The reason is that the first part (lines 57-78) is a part of the Introduction which sets up the scene for the paper. The second part (lines 100-206) is a part of the short history of glass as an art medium. If I moved the second part to the Introduction, the Introduction would be far too long.  If I moved the first part to the second part, the reader would not understand from the Introduction what the main drivers for writing the article are.

There are some gaps in this research; the conclusion is essentially accurate, but brief and somewhat one-sided. There are several well-known fine artists who work with glass without facilitation by Berengo, such as Larry Bell, Roni Horn and Olafur Eliason and, of course, Marcel Duchamp. They should be referred to. (Eliason is mentioned as exhibiting in Glasstress Gotika 2015, but he has been working with glass since the early 90s). While the article briefly touches on the crafts/fine art divide, there is no mention of the challenges Berengo studio  faces in facilitating artwork in a new (for the artists), and technically difficult, medium.

I appreciate the Reviewer’s comments and amended the text as required. Indeed, there are several well-known fine artists who work with glass and they use different facilitation of other fabricators or without facilitation (as Olafur Eliasson who founded his studio with the large professional team).

I have also added more about the challenges the Berengo studio faces in facilitating artwork of a new (for the artists), and technically difficult, medium.

65-70 not quite accurate: Harvey Littleton applied for and received funding to build a small furnace at the Toledo museum of art, not in his studio. There, he experimented with 5 of his graduate students. The author mentions this, without naming Littleton, in lines 190-192. This should be updated and brought together. https://www.cmog.org/article/harvey-k-littleton-and-american-studio-glass-movement

Thank you for the comments. I have updated the text as requested.

line 203-204: “the current President of the Corning Museum, Thomas Buechner” – This is  inaccurate: Thomas Buechner was the founding director of the Corning Museum of Glass, later president of Steuben Glass and the Rockwell Museum, but never president of the Corning Museum of glass. He died in 2010. https://www.cmog.org/bio/thomas-s-buechner

Again thank you for the comment. I have updated the text accordingly.

Reviewer 2 Report

The subject of the article is interesting; however the writing of the document is quite confusing.

The difference between point 1 and 2 should be better exemplified.

Now I will point out some ideas that need to be more detailed and explained.

  1. What is the artist that will represent Ireland in the Biennale? Is it Niamh O’Malley , do not put in parentheses. 

The introduction is confused. The author starts to present Bertil Valien as an example of an artist that worked at Ã…fors glass- 44 factory, not given a specific date. Although there are more examples of the relationship between glass art and industry also glass and research that are not mentioned.

Then the author goes back on time and writes about the 40’s and 50’s and then about  Czech artists and American Studio Glass movement.  

Then Adriano Berengo is not correctly introduce. Date of birth? Place?

  1. The author says that Glasstress is successfully continue until today, given the reference of the year 2015, a recent reference should be given.

Please rephrase the introduction

114-117 there is no reference on this paragraph

  1. Egidio Costantini- Date of birth? Place?

In 1. Short History of Glass as an art medium

There is no chronological order. From 13th century goes to 1950’s murano and back to 19th. This makes the article difficult to read.

  1. Vera Mukhina - Date of birth? Place?

What is the reason for choosing the artist Vera Mukhina over other artists ?

Please rephrase point 1.

In 2. Glass as a medium 

A better defined theoretical context is lacking.

In 3 Adriano Berengo and his projects to elevated importance of glass as art medium

Some of the artists are just mentioned. It would be important to do a more in-depth investigation, rather than merely reporting.

Why was it important for Louise Bourgeois or Mona Hatoum to make these pieces?

As it was mentioned with Ai Weiwei pieces, it would be more interesting to read case studies of other artists.

Author Response

I would like to thank for the Reviewer’s thoughtful comments that have allowed me to improve my article. Below are my replies (in red) to specific points raised. The modified parts of the article are indicated in blue.

The subject of the article is interesting; however the writing of the document is quite confusing.

The difference between point 1 and 2 should be better exemplified.

I have modified both sections to explain my ideas in a hopefully clearer way.

Now I will point out some ideas that need to be more detailed and explained.

22. What is the artist that will represent Ireland in the Biennale? Is it Niamh O’Malley , do not put in parentheses. 

The text has been corrected.

The introduction is confused. The author starts to present Bertil Valien as an example of an artist that worked at Åfors glass- 44 factory, not given a specific date.

I have added an explanation about Bertil Valien as an artist who developed skills to work with glass as a fine art medium. The date has been added.

 Although there are more examples of the relationship between glass art and industry also glass and research that are not mentioned.

I have added more information about artists and industries, research and about different ways the artists cope with solving problems.

Then the author goes back on time and writes about the 40’s and 50’s and then about  Czech artists and American Studio Glass movement.

The reason I did that is because it was one of the most important developments for fine artists who want to work with glass.  

Then Adriano Berengo is not correctly introduce. Date of birth? Place?

I have modified the text as requested.

  1. The author says that Glasstress is successfully continue until today, given the reference of the year 2015, a recent reference should be given.

I have modified the text and explained how Glasstress evolved through time, until the present day, to support better the development of application of glass as a fine art medium

Please rephrase the introduction

114-117 there is no reference on this paragraph

The text has been modified as requested.

  1. Egidio Costantini- Date of birth? Place?

The text has been modified as requested.

In 1. Short History of Glass as an art medium

There is no chronological order. From 13th century goes to 1950’s murano and back to 19th. This makes the article difficult to read.

I have added information about history of Murano between 13th century to 1950’s. I have modified the text extensively to keep the chronology and improve clarity of explanations

  1. Vera Mukhina - Date of birth? Place?

The text has been modified as requested.

What is the reason for choosing the artist Vera Mukhina over other artists ?

I have added an explanation that Vera Mukhina was probably the first fine artist who started cooperation with technologist to be able to include glass in her artistic practice. Her glass sculpture were quite big, around 70, 80 centimetres tall.

Please rephrase point 1.

I have modified the text extensively to keep the chronology and improve the clarity of explanations.

In 2. Glass as a medium 

A better defined theoretical context is lacking.

I have modified the text 

In 3 Adriano Berengo and his projects to elevated importance of glass as art medium

Some of the artists are just mentioned. It would be important to do a more in-depth investigation, rather than merely reporting.

Why was it important for Louise Bourgeois or Mona Hatoum to make these pieces?

I have modified the text. I have added explanations why I included those particular artists and what caused Adriano Berengo to include them in his exhibitions.

As it was mentioned with Ai Weiwei pieces, it would be more interesting to read case studies of other artists.

Due to space constraints, I had to restrict the attention to just a few artists. I have paid particular attention to Ai Wei Wei because of his prominence and because he started to use glass regularly in his art and he could himself produce his pieces in the Berengo Studio.

Additionally, I have added more examples in the text how other fine artists use glass in different ways and how they solve the problems.

Round 2

Reviewer 2 Report

Some considerable changes were made on the article, however I would suggest to add the date of birth and place of Zdenek Lhotsky, line 239 and also explain what consist the group "The Stubborn Ones"- this information can be placed in footnote.

Line 251 - Olafur Eliasson is presented and in the paper says that he move to Berlin in 1995, there is no reference